# Perceptual Styles and Cannabis Consumption Prediction in Young People

**DOI:** 10.3390/ijerph17010288

**Published:** 2019-12-31

**Authors:** Carlos Herruzo, María J. Pino, Valentina Lucena, Javier Herruzo

**Affiliations:** Psychology Department, Facultad de Ciencias de la Educación, University of Cordoba, 14071 Córdoba, Spain; ed1piosm@uco.es (M.J.P.); ed1lujuv@uco.es (V.L.); ed1hecaf@uco.es (J.H.)

**Keywords:** risk perception patterns, cannabis use, information about drugs, prevention policies

## Abstract

Given that risk perception has been found to be both a vulnerability and a protective factor with respect to consumption, the objectives of this study were to find out whether there exist specific patterns of risk perception associated with cannabis use and, if so, how they relate to cannabis consumption and to the sources of information on drugs accessed by young people. An ex post facto study was carried out with 1851 young Andalusians aged 18 to 29, using an adaptation of the Andalusian Government “Andalusian Population versus Drugs” survey. For the first objective, a cluster analysis was carried out in which three perceptual style groups were formed: “Strict”, “Permissive-Awareness” and “Lax”. Cannabis use in the “lax” group was found to be 14.31 times more frequent than in the “strict” group and 2.75 times more frequent than in the “permissive-awareness” group. A logistic regression analysis was also performed, which correctly predicted 80.4% of users and non-consumers. Correlation was found between perceptual styles and the sources of information used about drugs. This study identified three different risk perception styles that heavily correlated to cannabis consumption and to the type of sources young Andalusians use to obtain information about drugs, suggesting the need for a change in preventive policy.

## 1. Introduction

Cannabis is the most widely used drug worldwide. The United Nations Office on Drugs and Crime (UNODC) Survey for 2019 (based on data from 2017) estimates that about 3.8% of the global population aged between 15 and 64 used cannabis at least once in 2017 [1]. Spain is the highest ranking country in Europe, and one of the highest in the world, in cannabis consumption. According to the 2019 Spanish Government Drugs Report [2], prevalence in the last year among young adults was 18.3%, the figure for men more than doubling that for women. In most European countries, recent survey results reveal that cannabis use in the past year has remained stable or is increasing among young adults [3].

The widespread use of cannabis is an important public health issue. The consequences of cannabis consumption are well documented, and there is a broad consensus regarding its short-term negative impact in areas like driving, work and academic performance, and even social and family interaction [4]. Chronic cannabis consumption has also been associated—although not with such a high degree of consensus—with certain cognitive dysfunctions in areas like perception, reaction time, learning, memory and concentration, which may lead to more serious damage affecting social skills and emotional control [5,6]. In this regard, it should be noted that some recent studies have shown significant links between cannabis use and the loss of up to eight intelligence quotient points [7], the onset of psychosis, [8,9], and also affective disorders, anxiety and the risk of suicide [9,10].

However, many people do not believe that regular cannabis consumption involves a health risk [11]. Among young adults in the USA, the percentage of people aged 18 to 25 who perceived a high risk in regular cannabis use in 2012 was 23.8, a figure considerably lower than that for 2002 (35.6) [12], and this figure has decreased continuously over the last decade [11]. In Europe, the figure stands at almost 20% [3]. Although knowledge of drugs and other health risks does not in itself predict preventive behavior or the avoidance of high-risk activity, we now know that risk perception regarding a substance—defined as “perception of the negative effects of drug consumption” [13] (p. 39)—deeply influences people’s decisions about taking drugs and seeking help [14]. The risk levels of different drugs, as perceived by young people, influence their consumption: the riskier the consumption of a particular drug is seen to be, the less it is consumed, and vice versa [15,16,17]. However, we also know that this perception changes among those who have actually consumed drugs. In this regard, Okaneku et al. [18] studied changes in the perception of risks associated with cannabis consumption and found that people who had consumed cannabis in the previous month reported lower risk perception for both regular and occasional consumption. Similarly, the structural equation model developed in Europe by Grevenstein, et al. [19] showed that risk perception significantly influences consumption patterns for substances like tobacco, alcohol and cannabis, and proved that changes in risk perception predict future changes in drug use. Moreover, cannabis consumption was also found to influence perception of the risk associated with that substance. Salloum et al. [20] observed how an increase in cannabis use was linked to a decline in risk perception and Blevins et al. [21] demonstrated a two-way relationship between risk perception and future cannabis use, with a stronger link between cannabis use and lower subsequent risk perception. In the same vein, Parker et al. [22] showed that the level of risk perception made it possible to predict consumption of cannabis over the next two years.

Ultimately, according to Grevenstein et al. [19], risk perception can be understood as a protection, or risk, factor, which also changes as consumption of the substance increases, because contact or experimentation with a substance produces a reappraisal of its risk. Their study, however, was unable to clarify the direction of the influence: i.e., whether the main effect is motivational (risk perception influences consumption patterns) or due to reappraisal (experimentation with the substance leads to changes in risk perception).

Given the importance of risk perception and considering that perceptions are subjective, often generalized phenomena, based on beliefs and attitudes [23], the question arises of whether perception of the risks associated with cannabis use corresponds to certain stereotypical perception patterns, and, if so, what those patterns are and the extent to which they are related to consumption of the drug. However, little research has been carried out in this area. To the best of our knowledge, studies into risk perception have mainly focused on the relationship between patterns of use and risk perception, or on the perceived danger ranking of different drugs, but not on the problem of perception patterns per se [24,25].

In this regard, the biggest differences in the perception of risks associated with drug consumption have been linked more with the type of drug involved than with consumption patterns. As a result, the highest risk perception corresponds to the habitual use of heroin, followed by hallucinogens, cocaine, and ecstasy. Next would be the consumption of tobacco. Less risk is attributed to the habitual consumption of hashish and tranquillizers and, at the bottom of the list, to the daily consumption of 5 or 6 alcoholic drinks [24,25,26,27,28]. In Spain, risk perception associated with the consumption of legal substances (alcohol, tobacco and hypnosedatives) is lower than that associated with illegal drugs like cannabis, cocaine and hallucinogens [26,29]. Herruzo et al. [30] noted that young people’s perception patterns vary for each drug. Alcohol-related health problems, for example, are considered less dangerous than those caused by cannabis, and cannabis-related problems are in turn considered less dangerous than those caused by tobacco. The same authors also noted differences between the sexes. In the same vein, Pino et al. [31] found that perception of the risk alcohol consumption posed regarding traffic accidents was higher than that corresponding to other drugs, and that this greater awareness (despite being associated with consumption) may have been due to campaigns and sanctions to discourage drunk driving. Therefore, there is a gap in research into risk perception that needs to be filled.

Finally, given that perception is a cognitive process dependent on each individual’s information about different things like contexts, other people, objects, etc. [23,32,33], and given that the context in which information is provided would be important for reinforcing or questioning the validity of a perception [23,34,35], the question arises of whether sources of information about drugs used by young adults are associated with their patterns of risk perception regarding cannabis consumption. As has been identified in the literature, risk perception depends on certain factors such as age, gender, substance use, frequency of use, having family members who use substances, level of education, level of education of parents and the composition of the family group [35,36,37,38] and most of these factors relate to sources of information and contact with drugs. However, very few studies have addressed this issue.

Considering, then, the proven consequences of cannabis consumption, the relationship between cannabis use and risk perception, and the corroboration of bias in the perception of risks associated with the consumption of certain substances, it is clearly necessary to study risk perception in greater depth. The first objective of this study is, therefore, to identify whether perceptual styles (perceptual patterns) actually exist. A second objective is to ascertain the extent to which they are related to cannabis consumption. Finally, the third objective focusses on the relationship between perceptual styles and the sources of information about drugs consulted by young people. This information will be useful when designing preventive campaigns or measures to reduce cannabis consumption.

## 2. Materials and Methods

### 2.1. Sample Group and Design

The sample group for this study comprised 1851 young Andalusians aged between 18 and 29, distributed by sex, age and occupation as shown in Table 1. The participants came from the provincial capital (18.6%) and from towns and villages, clustered by administrative district (81.4%). In total, 6.5% of them did not fully complete the questionnaire. The study was conducted using a prospective, ex post facto design and in accordance with the Declaration of Helsinki, and the protocol was approved by the Ethics Committee of Biomedical Research in Andalusia (Spain).

### 2.2. Procedure

Data were collected in collaboration with teachers at the University of Cordoba and coordinators from the “Ciudades ante las drogas” (“Cities verses Drugs”) program, who administered the questionnaire (described in the next section) to members of the young population, randomly selected from those attending training courses, places of work, gymnasiums and social centers, in their town or village. At the University of Cordoba, a representative sample was selected by clusters (classrooms). The teachers were contacted and set aside half an hour of their class time for the students to fill out the questionnaire. The personnel conducting the survey had been trained beforehand to be able to give full, clear instructions. The questionnaire’s instructions section stated that it was for a study into the consumption of different substances and certain behavior by young people, and that the objective was to identify preventive and corrective measures with which to address possible problems associated with such behavior. The instructions also emphasized the importance of participating in the survey, and stated that the data would be processed statistically in a manner that would guarantee total anonymity.

### 2.3. Instruments

The questionnaire used was that of Ruiz-Olivares et al. [26]. Its main objective is to explore substance (alcohol, cannabis, cocaine and synthetic drugs) consumption patterns, sociodemographic variables (like age, sex, university level education, family background, etc.), risk perception and non-substance addictions, and it does so by means of four blocks of questions: sociodemographic data, consumption patterns, risk perception and non-substance addictions. In this study, the last block was not used. The other three blocks included questions from the Andalusian versus Drugs survey X [28] (validated by the European Monitoring Centre for Drugs and Drug Addiction). The questions about consumption patterns were divided up to address each specific drug (alcohol, cannabis, cocaine and synthetic drugs) and had four answer options: “Never consumed”, “Consumed at some time in my life”, “Consumed in the last 12 months”, and “Consumed in the last 30 days”. For the risk perception block, the participants were asked to evaluate the different types of risk associated with the consumption of each different substance, such as the possibilities of having (a) traffic accident; (b) serious psychological or physical problems; (c) difficulties in interpersonal relationships; (d) legal problems (being arrested or fined, having their ID document confiscated, etc.); and (e) irreversible physical or mental health problems. A Likert-type scale of 1 to 5 was used (1 = “never causes problems”; 2 = “rarely”; 3 = “sometimes”; 4 = “causes quite a few problems” and 5 = “always causes problems”). Cronbach’s alpha coefficient for the risk perception block was = 0.967, classed as excellent by George and Mallery [39] (p. 231). For the consumption pattern and non-substance addiction blocks, the alpha coefficients were 0.777 and 0.836, respectively. This study used the questions about cannabis use and risk perception regarding cannabis. The specific alpha coefficient for the cannabis risk perception questions was 0.876. Sources of information about drugs were assessed by means of “yes” or “no” answers for each item on a list of 11 information sources, the participants being asked if they received information from the source in question.

### 2.4. Statistical Analyses

The information obtained in the questionnaires was fed into the SPSS 20.0 (IBM, Armonk, NY, USA) program to create a database. Since the literature contained no clues about the optimal number of groups to establish, a two-step cluster analysis was first conducted. This method automatically selects the number of clusters [40]. It found three to be the ideal number. SPSS classifies results as described in Kaufman and Rousseeuw [41]. In this case, three clusters were classified as “good”. A K-means cluster analysis was then conducted to group the subjects together according to similarities in their perception of the different risks associated with cannabis use. The people in each group were therefore as similar as possible in terms of risk perception, and at the same time different from those in the other two groups. The K-means method produces optimal results when the number of groups to be used is known [40]. A cluster analysis was carried out with all the answers for all six risk dimensions (traffic accident; serious psychological problems; difficulties in interpersonal relationships; legal problems—being arrested or fined or having one’s ID document confiscated, etc.; reduced ability to perform tasks; and irreversible health problems). An ANOVA test was then conducted to determine the differences between clusters, and a Chi Square test was conducted to determine differences by sex and age. To explore the relationship between risk perception and cannabis consumption, the Odds Ratio (OR) for “Used in the last 30 days” was calculated between each pair of clusters. To explore the relationship between risk perception style and cannabis consumption, a binary logistic regression analysis was later performed, using the results of the cluster analysis as an independent variable and dichotomized cannabis consumption as a dependent variable, selecting only cases of non-consumption (=0) and consumption over the past 30 days (=1), deleting the other two consumption categories to create the variable (61.8% of the sample was conserved and 38.36% deleted). For the third objective, a Chi Square test was conducted among clusters in each information source.

## 3. Results

The first objective in this study was to discover whether or not common risk perception patterns exist. The grouping of risk perception answers in the sample was therefore closely analyzed. Figure 1 shows the results of the K-means cluster analysis for the three groups (three having been indicated by the hierarchical cluster analysis as the optimal number of clusters). The highest scores, reaching almost 5 (between 4.68 and 4.92), represent the “Always causes problems” answer, and can be called “strict” perceivers. This cluster represented 53.2% of the sample. The second group, with scores around 2 (“Rarely causes problems”) was denominated “lax”, indicating a more offhand type of risk perception. This group represented 15.3% of the sample. The third group was characterized by scores between 3.21 and 4.09. This cluster was denominated “permissive-aware” and represents 31.4% of the sample. The ANOVA revealed significant differences between clusters for all risk perception values (all *p* < 0.001). There were significant differences by sex (Chi square = 27.76; *p* < 0.001) but not by age (Chi square = 12.04; *p* = 0.061).

To explore the relationship between risk perception and cannabis consumption (the second objective), the Odds Ratio (OR) was calculated between the “strict” and “lax” clusters for cannabis consumption. The result was 14.31, meaning that cannabis consumption during the previous 30 days was 14.31 times more frequent among young people in the cluster with a lax perception of cannabis-related risk than among those in the cluster with stricter risk perception. The OR between the “permissive-aware” cluster and the “lax” cluster was 2.75, while between the “strict” and “permissive-aware” clusters, it was 5.20. To characterize the relationship between risk perception style and cannabis consumption, a binary logistic regression analysis was also performed, using a new variable, calculated by dichotomizing cannabis consumption, as the dependent variable, and membership of the prototype perception group (“Strict”, “Permissive-Aware” or “Lax”) as the independent variable. This method made it possible to predict a subject’s risk of cannabis consumption from their membership of a specific perceptual style group. This variable (perceptual style) alone correctly predicted 80.4% of the subjects (90.2% of non-consumers and 41.5% of consumers; 95% confidence interval for EXP (B) = 0.266; Lower CI = 0.216; Upper CI = 0.327).

Finally, considering the relationship between perceptual phenomena and the drug information sources used by young people, and in order to obtain information useful for the planning of preventive actions, we explored the relationship between the main drug information sources accessed by young people and both perceptual style and consumption (the third objective).

Figure 2 shows the information sources according to perceptual style. Note the infrequency of recourse to official sources and health sector professionals to obtain information. In contrast, the most frequently used source is the mass media. Surprisingly, however, the “lax” group and the “strict” group differed regarding the use of information obtained from friends or people in contact with drugs, which occurs more frequently in the “lax” group (the group associated with higher consumption levels). In contrast, the information source most frequently used by those in the “strict” group is parents.

## 4. Discussion

The main objective in this study was to establish the existence of risk perception patterns among young people between the ages of 18 and 29, a topic scarcely studied, if at all, in the literature. Cluster analysis produced three risk perception groups characterized by laxity, permissive awareness and a stricter attitude towards the dangers associated with cannabis consumption. Those in the “strict” group, representing over 50% of the sample, believe that cannabis always causes problems associated with traffic accidents, health, the ability to perform certain tasks, relationships with other people and/or legal infringements. People in the “permissive-aware” cluster typically showed perceptions of risk lower than those in the “strict” group, varying between “sometimes” and “causes quite a few problems”. One of the most revealing results was that 15.3% of young people have a lax perceptual style: they broadly believe that cannabis rarely causes problems. This is important because, as mentioned at the beginning of the article, the consequences of cannabis consumption are well documented. The discovery that around 1 in 6 young people do not consider cannabis risky shows that they are not aware of its effect on driving, work and academic performance, IQ loss, or the increased risk of psychosis, or, on the other hand, that early contact with cannabis may possibly induce a laxer, more offhand risk perception more compatible with its consumption. This, however, needs to be studied in greater depth.

Our results are coherent with some authors [16,19,24,25] who found a higher risk perception among girls than boys and a higher prevalence of cannabis use among low-risk perceivers. They are also congruent with those of Merrill [42], who found that U.S. students between the ages of 11 and 18 who did not consider marijuana harmful were nine times more likely to have consumed it. The young Andalusians in the perception cluster we called “strict” were 14.3 times less likely to consume cannabis than those in the group we called “lax”. This confirms that those identified as having a high risk of consuming cannabis were also the ones most likely to consider cannabis less harmful. Moreover, if the binary logistic regression results are considered, this means that the perceptual style variable (“lax”, “permissive-aware”, “strict”) is by itself able to correctly allocate 80.4% of subjects to the consumption/non-consumption group, and it is therefore possible to use perceptual style to predict cannabis use. Although these results should be taken with caution since the percentage of users correctly identified was 41.5% and that of non-users 90.2%, this can facilitate the identification of risk groups insofar that young people are more likely to be honest if asked about their beliefs rather than about their consumer behavior.

Ideally, further research should be done to find out whether the model’s prediction would be more accurate if it included variables like the onset age for consumption of alcohol, tobacco and other drugs, or if it asked about regular use as opposed to occasional or infrequent use, but this aspect lies beyond the scope of this study.

The results obtained suggest a need for change in our prevention policy, justifying greater attention to the onset of consumption, usually during adolescence (67.4% of consumers start before the age of 18). Although, since this is a transverse study, they do not throw light on whether it is perception that influences consumption or vice versa, the results of Salloum et al. [20], Blevins et al. [21] and Parker et al. [22] demonstrating a two-way relationship between risk perception and future cannabis use and vice versa, and the fact that people in the “lax” group are 14.31 times more likely to be consumers than those in the “strict” group suggest that people with “stricter” risk perception will probably be the ones who avoid consumption while those who experiment with the substance from an early age will probably develop a “lax” pattern of risk perception. The fact that we found no differences by age probably suggests a trend of biographical perceptual stability coherent with the fact that experience with the substance influences risk perception more than other variables.

Regardless of the sequence in which perception and experience influence each other, the practical implications of this study for prevention policy seem undeniable, at least for Spain, one of the countries in the world where cannabis consumption is most prevalent. As noted by Grevenstein et al. [19], the age at which people start using cannabis remains unaltered after decades of research and information campaigns. This suggests that the measures adopted to prevent consumption have not been effective. In this regard, the results obtained about information sources used by young people take on particular importance. The fact that young people in the “lax” perception group source information more from friends, people who have been in contact with drugs, and internet, whereas those in the “strict” perception group make significantly less use of those sources (turning to parents rather than friends, for example), tells us that a hefty 20% of our young people receive information via channels which probably confirm their “lax” perception of cannabis as a harmless drug and, in turn, are likely to become impervious to information from the mass media, talks in schools and colleges, etc. This concurs with Montanari et al. [14], who affirmed that perception of cannabis-related risks influences requests for help. The challenge this study poses, therefore, is the need to take preventive action very early, not only transmitting information but also implementing initiatives which, taking into account the variables that influence early contact with cannabis, encourage people to heed such preventive instructions in real situations of risk [43]. Essentially, the aim should be to compete with the information proffered by peers and friends who have had contact with drugs by means of campaigns featuring those same sources—peers and drug users—but providing accurate information. The message should not be merely prohibitive, but should focus on facts about the consequences of cannabis consumption at an early age in areas important to the target audience, like mental illness, as established in reviews [5,8,9]. If this is done via internet and social networks—the other most commonly accessed source—it will probably be possible to establish a contrast with the “lax”, carefree attitude projected by peers who use drugs.

Finally, mention should be made of this study’s limitations. Some of these were inherent in its methodological design. For example, with respect to the inclusion of university students in the sample, data collection in the classroom inevitably involves some degree of selection bias and the results may be underestimated, because students who regularly attend classes are those with the healthiest lifestyles [26]. An attempt was made to reduce this bias by collecting data in the first two weeks of the semester, when attendance is highest.

Another limitation of our study was that the questions in the survey did not ask about regular cannabis use as opposed to occasional or infrequent use. Neither did we assess whether participants had ever been in trouble because of cannabis previously, or how this might affect risk perception. With a view to future research, as well as these aspects, it would also be interesting to consider whether participants are consumers of more than one substance. Moreover, as the answers to the questions in the questionnaire probably reflect attitudes more than anything else, it would also be interesting to explore relationships with personality variables.

## 5. Conclusions

In conclusion, this study identified three different risk perception styles: “lax” (15.3%), “permissive-awareness” (31.4%) and “strict” (53.2%). These styles heavily correlate to cannabis consumption, with consumption being 14.31 times more frequent in the “lax” group than in the “strict” group and 2.75 times more frequent in the “lax” group than in the “permissive-awareness” group. Moreover, perceptual style was able correctly to predict cannabis consumption/non-consumption for 80.4% of the subjects. Perceptual styles were also shown to correlate with the type of sources young Andalusians use to obtain information about drugs. Bearing in mind that young people are likely to be more sincere in reporting their opinion about drugs than their consumption habits, perceptual style may therefore be a useful tool for identifying target populations for preventive campaigns which, in the light of recent reviews [5,8,9,10] and the results of this study, should focus more on making people aware of the mental health consequences (psychosis) and the cognitive losses associated with cannabis use.

## Figures and Tables

**Figure 1 ijerph-17-00288-f001:**
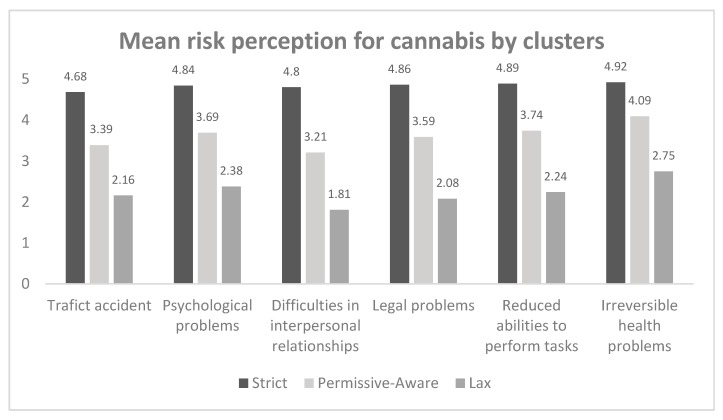
Mean risk perception on the Likert scale (1 to 5) for cannabis in six dimensions (*x*-axis) and for the three perceptual style groups produced by the cluster analysis. In the ANOVA, all differences were significant (*p* < 0.001) throughout the six dimensions.

**Figure 2 ijerph-17-00288-f002:**
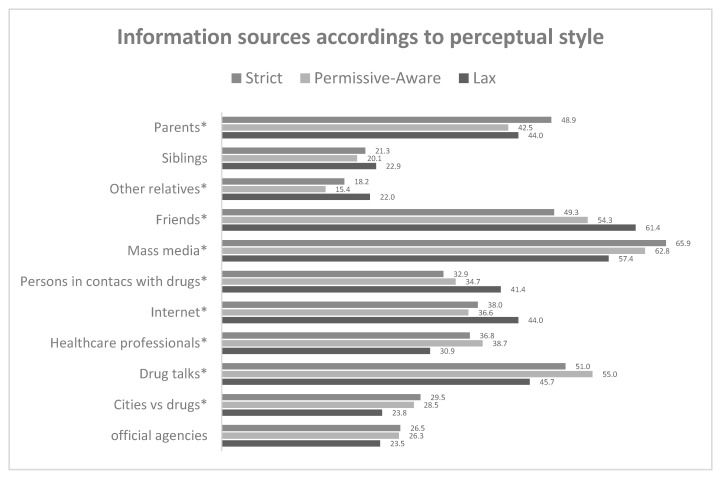
Most frequent information sources from which participants received information about drugs according to perceptual style. The asterisk (*) indicates that there were significant differences in the Chi Square Test (*p* < 0.05).

**Table 1 ijerph-17-00288-t001:** Sociodemographic characteristics of the sample.

*N* = 1851		Percentage
Sex	Women	52.6%
	Men	47.4%
Age	18 to 20	32.4%
	21 to 23	25.6%
	24 to 26	20.2%
	27 or older	21.8%
Occupation	Students	43.1%
	Workers	30.9%
	Students and workers (weekend)	13.8%
	Unemployed	11.4%
	No Response/Don’t Know	0.8%

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
