# Peer review of "Perceptual Styles and Cannabis Consumption Prediction in Young People"

_ijerph, 2019, doi:10.3390/ijerph17010288_

Round 1

Reviewer 1 Report

Very interesting work! Some of the graphs good have been done differently to show the differences more impressively. 
with the current trend of legalization, this paper demonstrates nicely how many people misperceive the risk of marijuana consumption .

Author Response

Dear reviewer, thank you so much for reading the article.

Reviewer 2 Report

This manuscript describes the investigation into the perception of risk of using cannabis and how this correlates with cannabis consumption and sources of information regarding risk among a cohort of young adults in Spain. The authors found that half of respondents were classified as "strict", whereas 15% were considered "lax" in terms of risk perception. They found significant correlations between sources of information and risk perception, as well as cannabis consumption. As this study only looked at a particular cohort - Andalusian young people - it is questionable whether these results can be generalized across other populations. For example, the Monitoring the Future study shows that substance use is trending downward among high school students in the United States, which conflicts with the statement in line 52: "drug use today remainss constant, or is even increasing."

Some reactionary language and interpretations are in need of reflection. First, over 2/3 of the respondents were above the age of 21, and therefore their experiences and objective risk of developing long-lasting medical problems are not expected to align with those of teenagers. Second, over 50% of the respondents were in the "strict" risk perception group; this is already a fairly substantial proportion of the population, and it is perhaps unrealistic to make more young people aware of the "dangers" of cannabis than they already are.

The conversation in lines 223-226 ("the discovery that around 1 in 6 young people do not consider cannabis dangerous shows that they are not receiving information properly") is overblown and reveals where the survey instrument is biased toward a strict perception. Are all young people supposed to fear cannabis? As an example, "the risk of having difficulties with interpersonal relationships" is probably objectively minimal outside of the most problematic cannabis consumers. The same argument can be lobbed at just about any of the questions asked, with the exception of "reduced ability to perform cognitive tasks." The reviewer - a cannabis pharmacology expert who is hardly a high consumer - shudders to think if they might be identified by this survey instrument as a likely pot smoker!

Since the survey instrument was not worded specifically, one must presume that the participants assumed "none/some/all of the time" to mean one's experience from a routine amount of cannabis consumed, or for an average consumer. Especially for participants who have experience with cannabis - "I smoked pot once and nothing bad happened" - it cannot be surprising that those participants would report a lax attitude toward risk.

The results of the binary logistic regression are not as clear-cut as described: whereas 90% of the non-consumers were correctly identified, only 41% of consumers were identified, suggesting that there was a large amount of false-negatives. This should be addressed.

Reviewer 3 Report

This study was designed to assess risk perceptions associated with cannabis use and its relationship with cannabis consumption and sources of information on drugs utilized by young adults. Based on survey results from 1,851 Andalusian young adults between the ages of 18-29, three perceptual styles emerged from a cluster analysis, including strict, permissive-awareness and lax. Cannabis use was higher among the lax and permissive-awareness styles relative to the strict style. Moreover, there were associations between the sources of information used by young adults and their perceptual styles. Although the overall premise of this study is strong, there are several issues that need to be addressed:

Introduction

-Although the introduction highlights literature that is directly relevant to the study (e.g., cannabis risk perceptions), the authors should consider reorganizing the presentation of the material. The introduction is very difficult to follow. For instance, given that risk perceptions are the major focus of the study, the literature in this area should have been covered earlier on in the introduction. Moreover, the introduction mentions studies and prevalence rates of drug use among adolescents, but not young adults. Given that the study focuses specifically on young adults, there should be more attention given to the prevalence of cannabis use among young adults and the risk perception literature among young adults.

-Related to the comment above, one of the study objectives is to explore the relationship between perceptual style and drug information sources used by young people. However, there is no rationale for why the source of drug information was included in the study. The authors should cite previous studies and/or provide a stronger rationale for the inclusion of this objective.

-The manuscript would be easier to follow if there were three objectives clearly listed, with one focusing on identifying the perceptual styles, one focusing on the relationship between these styles and cannabis consumption and the other focusing on the relationship between the styles and sources of drug information.

-At the top of page 2--what does EDADES stand for?

Materials and Methods

-The demographic information presented in the “sample group and design” subsection should be presented in a table.

-The procedure section should precede the instruments section to provide more context for the instruments described.

-The instruments section is difficult to follow. There is a description included for the items on the overall survey, but no discussion of how these items were used for the current study. For instance, based on the results, it looks as if participants were categorized into two groups, “has never consumed” and “consumed in the last 30 days.” If this is true, what happen to the individuals who endorsed “consumed at some time in my life” and “consumed within the last 12 months?” Were they deleted from the analyses? The authors will need to clearly state the outcomes in the current study and how the variables were defined for the purposes of the study.

-How consistent are the questions in the survey with those of existing reliable and valid measures of risk perception, cannabis consumption patterns and sources of drug information? In other words, are the measures/questions in the current study reliable and valid?

-Related to the comment above, the risk perception questions do not seem to accurately capture risks related to cannabis use. For instance, perceptions might be different if the questions focused on regular use versus occasional or infrequent use. This issue was not considered in the survey items. Also, there is likely other relevant areas in which individuals might perceive risk from cannabis aside from those included in the survey.

-The procedures section seems incomplete. The authors stated that the survey was administered to young people in their town or village randomly selected from those attending training courses, places of work, etc. However, in the Discussion section, the authors noted that one of the limitations of the study was that the data was collected in the classroom. The classroom setting was never mentioned in the Procedures section. Where exactly was the survey administered?

-Additional information is needed about the overall survey. For instance, were there any inclusion/exclusion criteria? How many individuals were ineligible? Why were they ineligible? Did anyone not complete the survey?

-The statistical analyses section needs more detail. A description of the cluster analysis, especially in relation to the current study, is needed to provide more context for the findings. After reading the analysis plan and the results, it is unclear how the three clusters were selected and what other issues were considered during the process. This is very important, especially if there are already reliable and valid measures and perception styles that have already been identified in the literature. A full description will allow for a stronger comparison between what is already available and the results from the current study.

-The analysis plan does not address the analyses for assessing the relationship between perceptual styles and cannabis consumption patterns and sources of drug information used by young adults. The results section briefly mentioned binary logistic regression, but this information belongs in the Methods section with the rest of the analytic plan.

-What about covariates (e.g., sex, age)? This is especially true given that the authors mentioned sex differences when discussing the literature in the introduction.

-The authors should condense the first two paragraphs on page 5 into one, as they both seem to cover most of the same information.

-Related to the comment above, the focus on perceptual styles and source of drug information seems out of place in the current study. If there is a strong rationale and/or current literature to support this objective, the authors should add this information to the introduction.

Discussion

-In the first sentence of the discussion section, the authors noted “the main objective in this study was to….., a topic scarcely studied in Spanish populations. However, this was not clearly discussed or highlighted in the Introduction. This gap in the literature should be mentioned in the introduction to help lay the foundation for the focus on Spanish populations in the current study.

-Again, the authors cite risk perception literature among adolescents in the discussion (e.g., Merrill). Is there any cannabis risk perception literature among young adults? If so, how do the current findings compare? What does this study add?

-Overall, the premise of the study is strong, but the lack of important details in all sections of the manuscript tempers enthusiasm and impact of the study.

Round 2

Reviewer 2 Report

I am comfortable that the edits made have addressed the concerns. 

Author Response

Thank you very much for your revisions.

Reviewer 3 Report

Overall, the authors were responsive to the reviewers' comments. However, there are still a few issues that need to be addressed:

-In the introduction, the authors should provide rationale for focusing on sources of information. The authors state that "the question arises of whether sources of information about drugs are associated with patterns of risk perception regarding cannabis consumption." However, there is no cited literature to support why this issue should be studied.

-In the introduction, the authors describe the high risk perceptions associated with other drugs, such as heroin and cocaine. Then, the authors discuss how risk perceptions are low with other drugs, such as marijuana. Given that the focus of this study is on marijuana, the authors should consider covering more literature on the risk perceptions specifically related to marijuana. This would help provide more context for the reader.

-Related to the comment above, a focus on risk perception marijuana literature would help to provide context for the overall study. For instance, how have other marijuana studies in particular defined risk perception? Have other marijuana studies identified perceptual styles related to marijuana? If not, then there needs to be more discussion in the introduction regarding how this study specifically contributes to the literature in this area.

-The authors still need to clarify how the consumption variable was created. Are the authors suggesting that they combined the three consumer categories (i.e., consumed at some time in my life, consumed in the last 12 months and consumed in the last 30 days) and compared it against those who endorsed "never consumed." If this is the case, this needs to be clearly stated. However, this does not appear to be the case because the authors then mention on page 4 that they only focused on consumption in the past 30 days versus no consumption at all. If this is the case, the authors need to clearly state that they deleted the other two consumption categories to create the variable.

-Related to the comment above, how did the deletion of two consumption categories affect the sample size?

-Minor comment, but periods are needed in the figures rather than commas.

-Overall, there is still some confusion regarding the design of the study. The authors should consider clearly describing all of the variables (e.g., consumption) in the study and providing rationale for their decisions (e.g., focus on sources of information). These details would make the study easier to follow.

Author Response

We would like to thanks to reviewer his or her valuable and helpful comments about the article. We have reproduced the reviewer comments (with a number) and then answered (after R:).

We hope these answers can satisfy the reviewer. We are at your disposal to any further comments you may want.

1-In the introduction, the authors should provide rationale for focusing on sources of information. The authors state that "the question arises of whether sources of information about drugs are associated with patterns of risk perception regarding cannabis consumption." However, there is no cited literature to support why this issue should be studied.

  R: We agree. We have rewritten the paragraph and added literature to support why this issue should be studied

2-In the introduction, the authors describe the high risk perceptions associated with other drugs, such as heroin and cocaine. Then, the authors discuss how risk perceptions are low with other drugs, such as marijuana. Given that the focus of this study is on marijuana, the authors should consider covering more literature on the risk perceptions specifically related to marijuana. This would help provide more context for the reader.

R: We agree. The introduction is focused on cannabis. We have specified more clearly in the text that the results of literature are specifically on cannabis risk perception (for example in line 68) to remark that these states are referred to cannabis.

3-Related to the comment above, a focus on risk perception marijuana literature would help to provide context for the overall study. For instance, how have other marijuana studies in particular defined risk perception?

R: To the best of our knowledge, there is no specific definition of marijuana risk perception in the literature different from the general one applied to cannabis.

4- Have other marijuana studies identified perceptual styles related to marijuana? If not, then there needs to be more discussion in the introduction regarding how this study specifically contributes to the literature in this area. 

R: We agree. We have introduced one statement about this aspect (lines 97-98)

5-The authors still need to clarify how the consumption variable was created. Are the authors suggesting that they combined the three consumer categories (i.e., consumed at some time in my life, consumed in the last 12 months and consumed in the last 30 days) and compared it against those who endorsed "never consumed." If this is the case, this needs to be clearly stated. However, this does not appear to be the case because the authors then mention on page 4 that they only focused on consumption in the past 30 days versus no consumption at all. If this is the case, the authors need to clearly state that they deleted the other two consumption categories to create the variable.

R: We have rewritten the paragraph in order to comply this suggestion as follow: “To explore the relationship between risk perception style and cannabis consumption, a binary logistic regression analysis was later performed, using the results of the cluster analysis as an independent variable and dichotomized cannabis consumption as a dependent variable, [and] selecting only cases of non-consumption (=0) and consumption over the past 30 days (=1), deleting the other two consumption categories to create the variable (61.8% of the sample was conserved and 38.36% deleted).”

We have deleted “and”  in line 187 and have added “deleting the other two consumption categories to create the variable (61.8% of the sample was conserved and 38.36% deleted).”

6-Related to the comment above, how did the deletion of two consumption categories affect the sample size? Included in the previous comment.

7-Minor comment, but periods are needed in the figures rather than commas.

R: We agree. Done

8-Overall, there is still some confusion regarding the design of the study. The authors should consider clearly describing all of the variables (e.g., consumption) in the study and providing rationale for their decisions (e.g., focus on sources of information). These details would make the study easier to follow.

R: We have attempted to comply with this suggestion. In lines 164-166 we have added a sentence about the assessment of source of information. In line 125 is indicated the design of the study.